# Pigment Production by *Paracoccus* spp. Strains through Submerged Fermentation of Valorized Lignocellulosic Wastes

Weronika Pyter [1], Jasneet Grewal [1] , Dariusz Bartosik [2] , Lukasz Drewniak [2] and Kumar Pranaw [1,*]

1   Department of Environmental Microbiology and Biotechnology, Institute of Microbiology, Faculty of Biology, University of Warsaw, Miecznikowa 1, 02-096 Warsaw, Poland
2   Department of Bacterial Genetics, Institute of Microbiology, Faculty of Biology, University of Warsaw, Miecznikowa 1, 02-096 Warsaw, Poland
*   Correspondence: k.pranaw@uw.edu.pl or kpranaw@gmail.com; Tel.: +48-579722792

**Abstract:** Due to the increasing emphasis on the circular economy, research in recent years has focused on the feasibility of using biomass as an alternative energy source. Plant biomass is a potential substitute for countering the dependence on depleting fossil-derived energy sources and chemicals. However, in particular, lignocellulosic waste materials are complex and recalcitrant structures that require effective pretreatment and enzymatic saccharification to release the desired saccharides, which can be further fermented into a plethora of value-added products. In this context, pigment production from waste hydrolysates is a viable ecological approach to producing safe and natural colorings, which are otherwise produced via chemical synthesis and raise health concerns. The present study aims to evaluate two such abundant lignocellulosic wastes, i.e., wheat straw and pinewood sawdust as low-cost feedstocks for carotenoid production with *Paracoccus* strains. An alkali pretreatment approach, followed by enzymatic saccharification using an indigenous lab-isolated fungal hydrolase, was found to be effective for the release of fermentable sugars from both substrates. The fermentation of the pretreated sawdust hydrolysate by *Paracoccus aminophilus* CRT1 and *Paracoccus kondratievae* CRT2 resulted in the highest carotenoid production, 631.33 and 758.82 µg/g dry mass, respectively. Thus, the preliminary but informative research findings of the present work exhibit the potential for sustainable and economically feasible pigment production from lignocellulosic feedstocks after optimal process development on the pilot scale.

**Keywords:** lignocellulosic wastes; pretreatment; saccharification; fermentation; microbial pigments; carotenoids; *Paracoccus*





## 1. Introduction

Color has been a part of our everyday lives for centuries. It is established that ancient Egyptians in 1500 BC added natural colorful extracts and wine to candies to enhance their desirability [1]. Nowadays, pigments and dyes are commonly used in the food, cosmetic, pharmaceutical, and textile industries. By 2027, the market value of dyes and pigments is expected to reach USD 33.2–49.1 billion [2]. However, the majority of currently applied coloring agents are obtained through chemical synthesis. A study conducted by the Swedish Consumers' Association (Sveriges Konsumenter), a member of the European Consumer Organization (BEUC), revealed that 9 out of 10 surveyed consumers worried about chemicals' impact on themselves, future generations, and the environment [3]. Though the addition of pigments is commercially important, as consumers tend to choose a preferable product according to its appearance, some synthetic pigments may cause serious health problems, as they can cause allergies or be carcinogenic or mutagenic [4,5]. For this reason, focus in recent years has shifted in the direction of natural pigments that can be obtained from plants, some animals, bacteria, fungi, yeasts, and algae [6].

Pigments produced by microorganisms have been treated with the greatest attention, as they are more resistant to light and temperature than more common plant-derived

pigments. Microbial pigments are also known to have medicinal properties like anti-mutagenic, anti-inflammation, anti-cancer, anti-oxidant, anti-obesity, anti-diabetes, and anti-microbial qualities. This can be advantageous to companies that use natural pigments, as consumers are becoming more and more conscious about their exposure to chemicals and tend to choose products that are as natural as possible [7–9]. Carotenoid pigments due to their bright coloration, structural diversity, and bioactivity have garnered high nutritional and pharmacological interest with an estimated market value of USD 2 billion by 2027 [5]. However, chemical routes drive the synthesis of 80–90% of commercially available carotenoids. Hence, in sync with consumer acceptance, the market demand for natural carotenoids, particularly from microbial sources, is gaining attention. The genuses *Dietzia* and *Paracoccus* are reported to be attractive candidates due to their promising bioactivities, which make them highly desirable for nutraceutical or pharmaceutical applications [10,11]. Nevertheless, there are many challenges hampering the microbial production of pigments, and the expensive synthetic fermentative medium is one of the major impediments. In this context, an ideal scenario would be to optimize microbial pigment production and extraction, using cheap and eco-friendly substrates for fermentation. This could rebrand the industry into a more sustainable and responsible business.

Agricultural residues (corn, rice straw, wheat straw, pulp peel, succulent bagasse, etc.) and woody forestry feedstock (birch, spruces, eucalyptus, etc.) are considered attractive substrates for the production of commercially valuable chemicals and biofuels [12,13]. The annual abundance of ~$100 \times 10^8$ metric tons of lignocellulosic biomass makes it a major alternative energy source, driving a paradigm shift from a non-renewable to sustainable bioeconomy model with environmental protection [14,15]. In the European Union, the generation of agricultural residue from different crops is around 6.29 million tons, nearly 17% of worldwide agriculture residue [16]. In Poland, major agricultural crops are soybeans, potatoes, wheat, oats, rye, maize, and barley, through which a total of 261,012.77 and 17,928,650.46 tons of agriculture and wood residue, respectively, were generated in the year 2019 [17]. Therefore, the utilization of these lignocellulosic wastes for microbial pigment production can provide the dual benefits of waste management and the production of high-value products with good market acceptance. However, the lignocellulosic, highly recalcitrant structure constituted by cellulose (40–50%), hemicellulose (20–30%), and lignin (10–25%) necessitates the pretreatment of these residues before enzymatic hydrolysis for the release of reducing sugars, which are subsequently utilized by microbes for pigment production [2,18].

Recently, several reviews [2,18,19] have elaborately emphasized the utilization of various waste feedstocks such as sugarcane bagasse, cottonseed meal, rice husk, and groundnut cake as attractive and eco-friendly substrates for microbial pigment production. With the above background, the present study attempted to evaluate two highly abundant but not commonly or easily valorizable lignocellulosic residues, i.e., wheat straw and pinewood sawdust, for pigment production via bacteria of the genus *Paracoccus* (*Alphaproteobacteria*). To the best of the authors' knowledge, there are no reports on the use of wheat straw or sawdust hydrolysate as nutrient sources for pigment production via *Paracoccus* spp. This study also becomes pertinent in the context of providing insights into the suitability of lignocellulosic hydrolysates without any detoxification step for microbial assimilation, as commonly generated inhibitors from lignocellulosic degradation are known to exert toxicity on fermenting microbes. Therefore, the present preliminary investigative study aims to provide a prototype of the suitability of unconventional lignocellulosic hydrolysates as alternatives to conventional media ingredients, which can have significant economic and environmental implications for pigment production after performing scale-up studies.

## 2. Materials and Methods

### 2.1. Materials

All chemicals and media used in this study were purchased from Sigma-Aldrich unless otherwise stated. Pinewood sawdust and wheat straw were obtained from the local market,

and the same lot was used throughout the study. All the solvents used were of HPLC grade. All the other chemicals and media used were of analytical grade.

### 2.2. Microorganisms, Culture Media and Growth Conditions

The *P. aminophilus* CRT1 and *P. kondratievae* CRT2 used in this study for pigment production were obtained from the Department of Bacterial Genetics, Institute of Microbiology, Faculty of Biology, University of Warsaw [11]. The strains were cultured in LB (Luria Bertani) broth with a composition of tryptone, 10 g/L; yeast extract, 5 g/L; and sodium chloride, 10 g/L. The strains were maintained on LB agar plates at 4 °C and subcultured at monthly intervals. The mother culture grew in LB ($A_{600}$~1.011) for 48 h and was always prepared fresh, and it was used as an inoculum for further experimentation. The crude hydrolytic enzyme used for saccharification was obtained using our laboratory fungal isolate *Trichoderma* sp. The fungal strain was maintained on potato dextrose agar (PDA) plates at 4 °C and subcultured at monthly intervals.

### 2.3. Pretreatment of Lignocellulosic Feedstocks

Locally procured pinewood sawdust (SD) and wheat straw (WS) were subjected to air-drying and subsequently ground and screened with sieve shakers, followed by storage in sealed airtight bags at room temperature (22 °C). The chemical composition of untreated wheat straw (% *w/w*) was 32.0 ± 0.25 cellulose, 45.9 ± 0.3 hemicellulose, and 14.2 ± 0.5 lignin, whereas the untreated sawdust comprised 44.0 ± 0.8 cellulose, 27.3 ± 0.6 hemicellulose, and 23.2 ± 0.7 lignin, as determined by standard NREL procedures [20]. Both were subjected to alkali pretreatment with sodium hydroxide (NaOH) using the modified method of Sharma et al. [21]. Briefly, the pretreatment was performed for 1 h with 1.5% NaOH at 5% (*w/v*) solid loading of the respective biomasses. The pretreated samples were subjected to repeated washings with distilled water until the pH reached neutral. The pretreated feedstock obtained was air-dried and used for further studies. The sawdust was also subjected to another alkali pretreatment assisted by a microwave using a modified method described by Jin et al. [22]. The sawdust was suspended in 2.25% (*w/v*) $Ca(OH)_2$ solution at 5% (*w/v*) solid loading. The suspension was subjected to microwave pretreatment at 1000 W for 6 min. The pretreated sawdust was rinsed with distilled water until the pH reached neutral, followed by air-drying for further use.

### 2.4. Saccharification of Pretreated Lignocellulosic Feedstocks

Three different sets of pretreated feedstocks, i.e., NaOH-pretreated wheat straw, NaOH-pretreated sawdust, and microwave-assisted $Ca(OH)_2$-pretreated sawdust, were subjected to saccharification using a crude enzyme hydrolysate obtained from the laboratory-isolated fungus *Trichoderma* sp. according to the modified protocol of Sharma et al. [21]. The alkali-pretreated feedstock was appropriately diluted with sodium citrate buffer (0.05 M, pH 4.8) to achieve an enzyme loading of 25 FPU/g and a substrate loading of 1% (*w/v*). The saccharification was performed in screw-capped vials at 37 °C with constant shaking at 150 rpm for 72 h. The samples were withdrawn at regular intervals, and the released reducing sugars were quantified by the DNSA method from the supernatant [23]. In all the saccharification experiments, the release of the reducing sugar was also determined from three controls that were run simultaneously: (i) untreated feedstock with buffer only, (ii) alkali pretreated feedstock with buffer only, (iii) untreated feedstock with 25 FPU/g of enzyme.

### 2.5. Fermentation Conditions for Pigment Production

The hydrolysates obtained after the saccharification of pretreated sawdust and wheat straw were used as low-cost substrates for pigment production with *Paracoccus* cultures. The reducing sugar content of all the hydrolysates was set to 1 g/L and supplemented with 0.5% (*w/v*) yeast extract. The pH of fermentative media was adjusted to 7.0, and kanamycin (50 µg/mL) was added after sterilizing it using 0.22 µm membrane syringe

filters. The respective hydrolysate media were inoculated with a 4% (*v/v*) primary inoculum of *P. aminophilus* CRT1 and *P. kondratievae* CRT2 grown in LB medium for 48 h. For control studies, the same bacterial strains were inoculated in commercial LB medium. All the inoculated fermentative media were incubated at 30 °C at 120 rpm for 72 h. The whole flasks were harvested after 72 h for pigment estimation. The growth in every medium was also measured spectrophotometrically at 600 nm. The appropriately diluted aliquots from harvested samples were also plated on an agar-solidified LB medium to determine the purity of microbial growth in the fermentative medium.

### 2.6. Pigment Extraction and Assessment

The pigment extraction from *Paracoccus* spp. cultures after 72 h incubation was performed in the dark and at room temperature using the modified method of Maj et al. [11]. The harvested cultures were centrifuged at 8000 rpm for 10 min, and the supernatant was discarded. The obtained bacterial pellet was suspended in 10 mL of acetone–methanol (7:2 *v/v*) solution and incubated at −20 °C for 15 min. Further, the samples were subjected to sonication for 5 min at 40% amplitude and centrifuged at 10,000 rpm for 6 min. The absorbance of the extracted solution was read at 453 and 488 nm, and the total amount of carotenoids was calculated as described by Liaaen-Jensen and Jensen [24]:

$$C = \frac{D \times v \times f \times 10}{2500} \tag{1}$$

where C is total carotenoids, *D* is the optical density at respective wavelength, *v* is the total volume (mL), *f* is the dilution factor, and 2500 is an average extinction coefficient for carotenoids.

The amount of carotenoids produced was expressed in μg/g of the dry mass of the cell pellet, obtained after drying the cell pellet at 65 °C for 24 h.

All the experiments were performed in triplicate and the data presented are with mean ± standard deviation of replicates.

## 3. Results and Discussion

### 3.1. Morphological Characteristics of Paracoccus Strains

For the investigative study, two *Paracoccus* spp. strains were used—*P. aminophilus* CRT1 and *P. kondratievae* CRT2, carrying plasmid pCRT01, which contain the carotenoid synthesis gene locus *crt* from *Paracoccus marcusii* OS22 [11]. The strains are efficient producers of a range of carotenes and xanthophylls. Both strains grew well on LB agar medium and formed small, round, convex, protruding, wet colonies of an intense orange color (Figure 1).

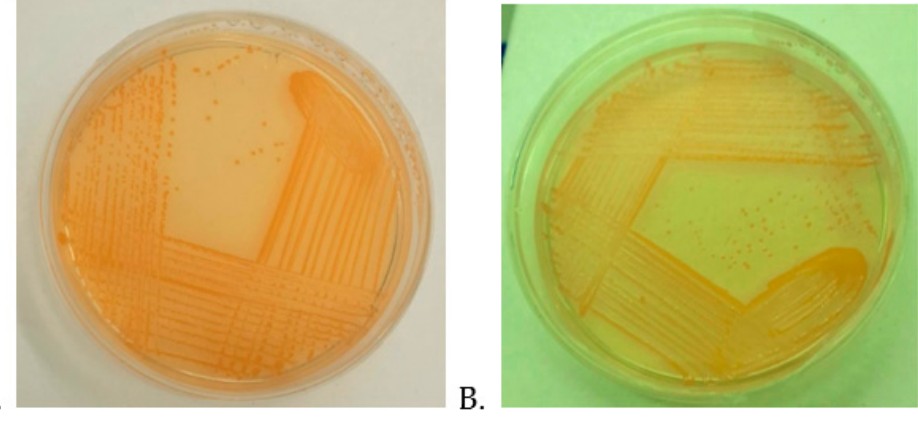

**Figure 1.** Colony morphology of (**A**) *P. aminophilus* CRT1 and (**B**) *P. kondratievae* CRT2 (LB agar).

### 3.2. Pretreatment of Lignocellulosic Feedstocks

The recalcitrant nature of lignocellulosic biomass is the decisive barrier to efficient enzymatic saccharification. Chemical (acid, alkali, or both) pretreatment mainly disrupts the lignin and hemicellulose binding within the cell wall, leading to a successive decrease in cellulose crystallinity. Chemical pretreatment firstly swells the substrate to increase its internal surface area and enables the removal of lignin molecules, which facilitate better enzymatic hydrolysis to yield reducing sugars [25,26].

Different lignocellulosic biomasses vary in their structural compositions, which necessitates an optimal pretreatment method with respect to their nature. In this study, an alkali pretreatment method was used for both lignocellulosic biomasses, i.e., wheat straw (WS) and pinewood sawdust (SD). As SD is known for its rigid and recalcitrant structure, another method, microwave-assisted alkali pretreatment with $Ca(OH)_2$, was also carried out. On the basis of enzymatic saccharification yield (Figures 2–4), it was concluded that both alkali pretreatment with NaOH and microwave-assisted alkali pretreatment with $Ca(OH)_2$ are promising pretreatment methodologies. Novakovic et al. [27] also reported that enzymatic hydrolysis improved after the alkali pretreatment of WS. Similarly, Sharma et al. [21] observed that alkali and acid pretreatment without steam sterilization was best suited for the pretreatment of corncob. In another study, Lu et al. [28] reported a significant increase of 80.78% fermentation yield after the alkaline pretreatment of birch sawdust compared to untreated, which is similar to the results obtained in the present study.

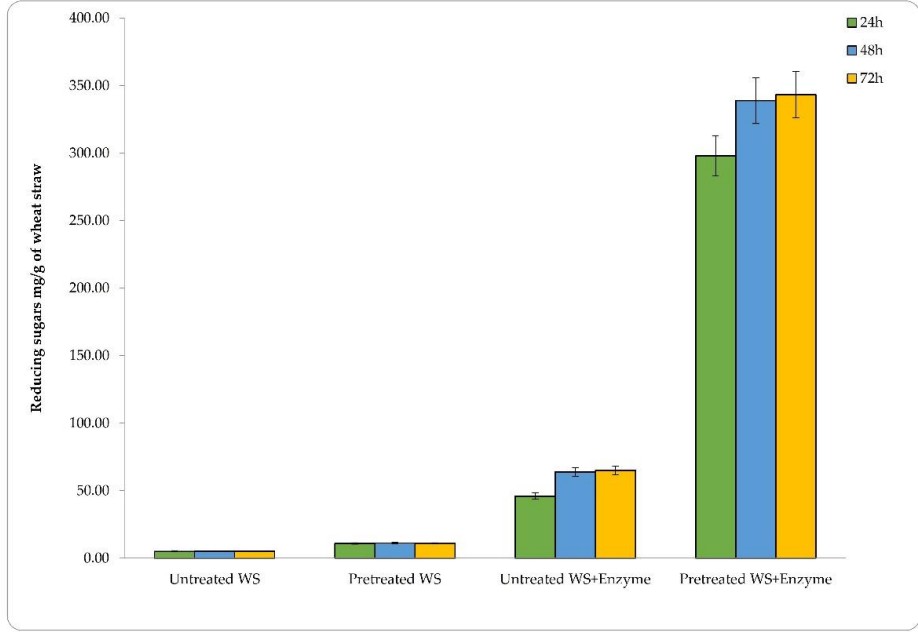

**Figure 2.** Reducing sugars obtained from untreated and NaOH-pretreated wheat straw, with or without the fungal crude enzyme.

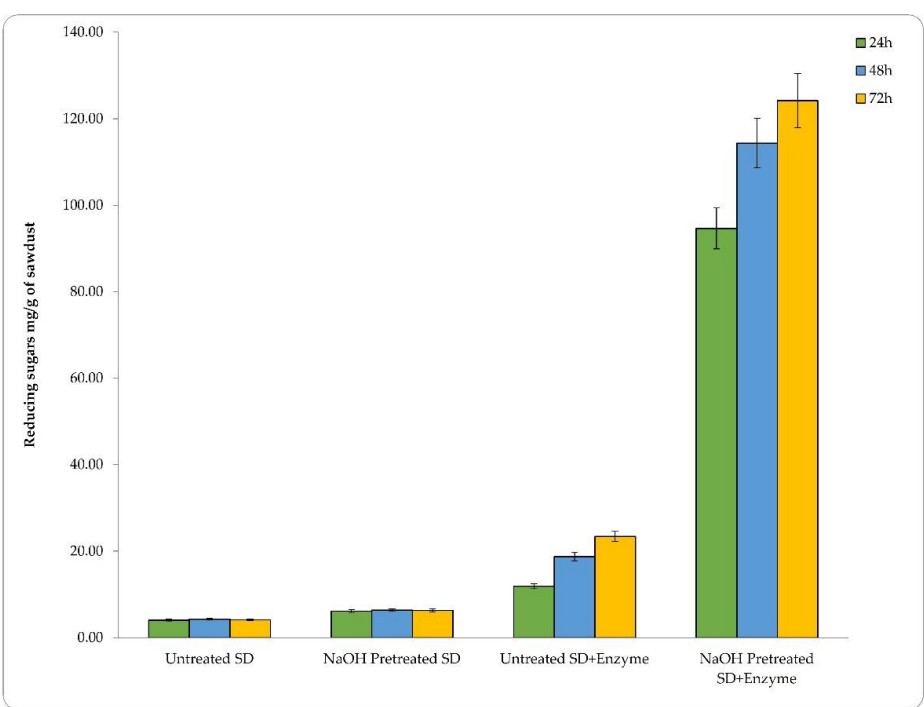

**Figure 3.** Reducing sugars obtained from untreated and NaOH-pretreated sawdust, with or without the fungal crude enzyme.

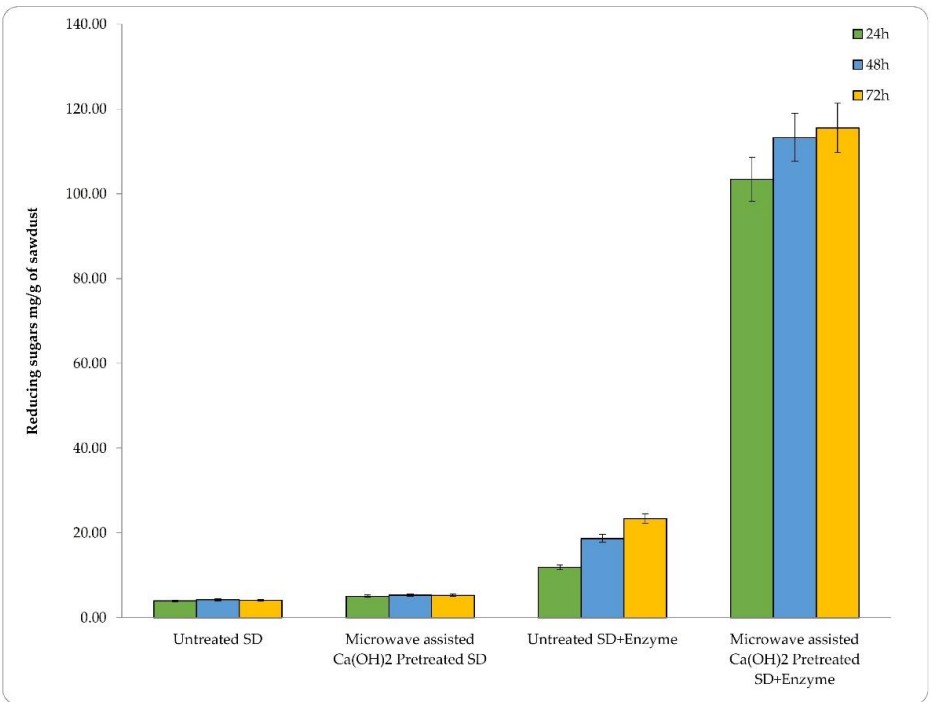

**Figure 4.** Reducing sugars obtained from untreated and Ca(OH)$_2$-pretreated sawdust, with or without the fungal crude enzyme.

### 3.3. Saccharification of Pretreated Lignocellulosic Feedstocks

Pretreatment and enzymatic saccharification are two vital steps toward increasing the yields of reducing sugars. The reducing sugar yield from the enzymatic hydrolysis of WS after different pretreatment strategies is shown in Figure 2. Untreated and pretreated materials without enzymatic saccharification were not able to release reducing sugars in

good amounts even after 72 h of incubation at 37 °C. The reducing sugar yield of the untreated WS without enzymatic saccharification was around 5.13 mg/g of WS, which was increased by more than two-fold to 11.03 mg/g of WS after NaOH pretreatment. The reducing sugar yield after the enzymatic hydrolysis of alkali-pretreated WS was attained as 343.26 mg/g of WS, which is almost 67% higher than the untreated control. In various studies, NaOH pretreatment, along with enzymatic saccharification, has also shown prodigious effects on sugar release [16,17].

Recently, Novakovic et al. [27] reported a saccharification yield ranging from 207 to 225 mg glucose/g of WS after the enzymatic saccharification of alkali-pretreated WS with the commercial enzyme cocktail Cellic Ctec2 for 96 h at 50 °C. In this context, the present study demonstrates a cost-effective approach with low energy inputs (37 °C operating temperature) to obtain a high reducing sugar yield (343.26 mg/g) using an indigenous fungal hydrolase for saccharification. Due to the more recalcitrant cell wall structure of SD, two different pretreatment techniques, such as NaOH pretreatment and microwave-assisted $Ca(OH)_2$ pretreatment, were used before enzymatic saccharification. Similar to WS, untreated SD and pretreated SD without enzymatic saccharification did not work well in terms of sugar release. Untreated SD without enzymatic saccharification resulted in the release of only 4.31 mg/g of SD reducing sugar, whereas the reducing sugar yield with NaOH and microwave-assisted $Ca(OH)_2$-pretreated SD without enzymatic saccharification was 6.31 and 5.31 mg/g of SD, respectively. The enzymatic saccharification of NaOH-pretreated SD led to the release of 124.17 mg/g of SD reducing sugar after 72 h (Figure 3). The microwave-assisted $Ca(OH)_2$ pretreatment technique was also comparable in pretreatment efficacy, though with a slightly lower yield: ~115.56 mg/g of SD (Figure 4). Nonetheless, this approach offered the advantages of simple operation and very short incubation time. Similarly, Jin et al. [22] also revealed the efficacy of microwave-assisted alkali pretreatment in enhancing the enzymatic saccharification yield from catalpa sawdust. The lower yield of reducing sugars in SD, as compared to WS, could be attributed to the different structural compositions of the two biomasses, as well as the higher lignin content in SD, which might generate a higher number of inhibitory compounds to hinder the saccharification step.

*3.4. Carotenoid Production by Paracoccus Strains via the Fermentation of Lignocellulosic Hydrolysates*

It is an established phenomenon that pretreatment procedures usually generate different adverse byproducts such as common aliphatic carboxylic acids, including acetic acid, formic acid, and levulinic acid, and furan aldehydes such as furfural and 5-hydroxymethylfurfural (HMF), which are not conducive for the subsequent steps of saccharification and fermentation. The amount of these inhibitory byproducts generated varies with the chemical structure of the lignocellulosic waste and the pretreatment reaction conditions but may require detoxification steps to prevent toxicity to the fermenting microbes [29,30]. Though the exact titers of these inhibitory derivatives were not determined due to constraints, the growth of the *Paracoccus* strains was monitored in saccharified feedstock hydrolysate to evaluate the suitability of these microbes for the valorization of lignocellulosic feedstocks without a detoxification step.

After 72 h of incubation, it was observed that the growth (in terms of optical density) of both *P. aminophilus* CRT1 and *P. kondratievae* CRT2 in their respective hydrolysates was comparable with the control medium, i.e., LB broth (i.e., $OD_{600} > 1.43$) (Supplementary Figures S1 and S2). *P. aminophilus* CRT1 exhibited luxuriant growth in NaOH and microwave-assisted $Ca(OH)_2$-pretreated SD hydrolysate medium, i.e., $OD_{600} > 1.81$ and $OD_{600} > 1.78$, respectively, while the growth of *P. kondratievae* CRT2 was also comparable with $OD_{600} > 1.65$ and $OD_{600} > 1.60$, respectively (Figure 5). This was in agreement with the previously reported capability of both strains, i.e., *P. aminophilus* and *P. kondratievae*, to grow on low-cost substrates such as M9 minimal medium and raw industrial effluent supplemented with molasses [11].

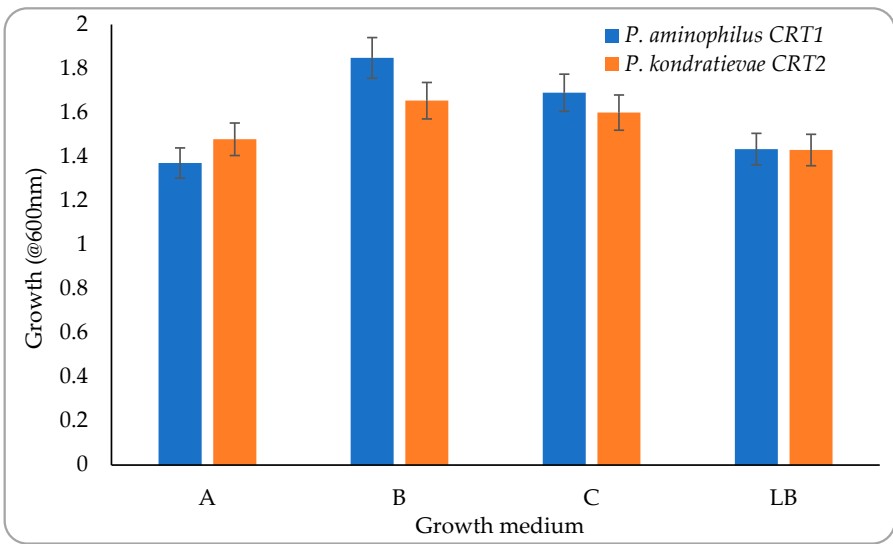

**Figure 5.** Growth of *P. aminophilus* CRT1 and *P. kondratievae* CRT2 in different media, i.e., A (based on NaOH pretreated wheat straw hydrolysate), B (based on NaOH-pretreated sawdust hydrolysate), C (based on microwave-assisted Ca(OH)$_2$-pretreated sawdust hydrolysate), and LB medium after 72 h of incubation.

As exhibited in Figure 6, the *P. aminophilus* CRT1 produced 677.13 μg/g of carotenoids in LB broth, which was comparable with 631.33 μg/g of carotenoids obtained in microwave-assisted Ca(OH)$_2$-pretreated sawdust hydrolysate-based medium. The efficiency of utilizing a low-cost carbon source for carotenoid production with *P. aminophilus* CRT1 is promising, as the yield was only 6.8% lower than commercial LB medium.

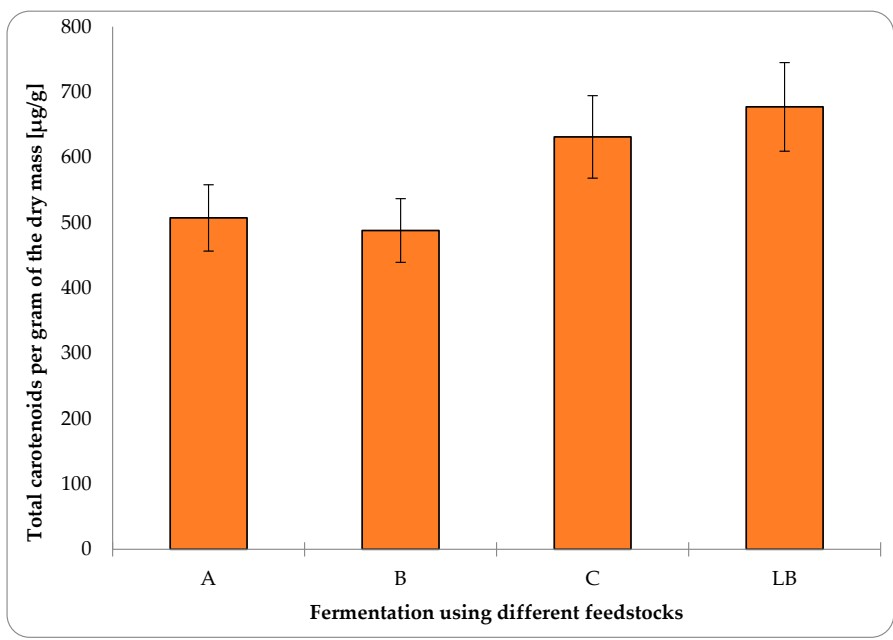

**Figure 6.** Total carotenoids achieved using *P. aminophilus* CRT1 cultivated in different media: A (based on NaOH-pretreated wheat straw hydrolysate), B (based on NaOH-pretreated sawdust hydrolysate), C (based on microwave-assisted Ca(OH)$_2$-pretreated sawdust hydrolysate), and LB (Luria Bertani broth medium) after 72 h of incubation.

In comparison, *P. kondratievae* CRT2 proved to be a more promising strain for the utilization of lignocellulosic feedstocks for pigment production (Figure 7). The highest carotenoid production (758.82 μg/g of dry mass) was observed when grown in the medium

containing NaOH-pretreated sawdust hydrolysate, which was even higher than the titer obtained in nutrient-rich LB medium (748.66 µg/g).The carotenoid production using NaOH-pretreated wheat straw hydrolysate and microwave-assisted Ca(OH)$_2$-pretreated sawdust hydrolysate was 467.40 and 600.75 µg/g of dry mass, respectively, which also supported the robustness of *P. kondratievae* CRT2 for utilizing the cost-effective lignocellulosic hydrolysates without any detoxification step.

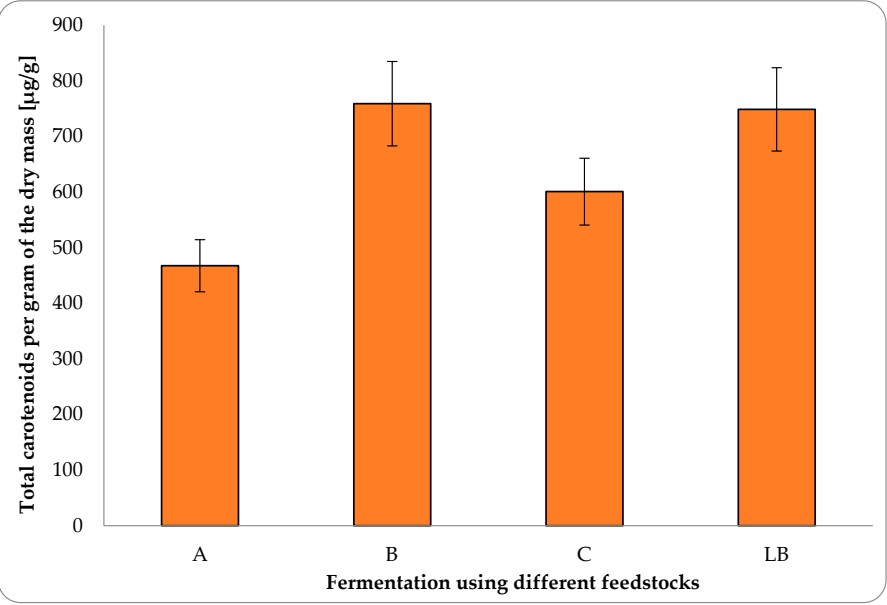

**Figure 7.** Total carotenoids achieved using *P. kondratievae* CRT2 cultivated in different media: A (based on NaOH-pretreated wheat straw hydrolysate), B (based on NaOH-pretreated sawdust hydrolysate), C (based on microwave-assisted Ca(OH)$_2$-pretreated sawdust hydrolysate), and LB medium after 72 h of incubation.

Studies evaluating waste utilization for pigment production using *Paracoccus* strains or its gene cluster are scanty. Nevertheless, the yields obtained in the present study were comparable to or higher than the commonly reported titers for *Paracoccus* strains in previous studies. The carotenoid production using flue gas desulfurization wastewater (FGD) supplemented with molasses as feedstock by *P. aminophilus* and *P. kondratievae* was reported as 117.68 and 60 µg/g of dry mass, respectively [11]. The utilization of glycerol, an industrial effluent, as a cost-effective substrate by *Paracoccus* sp. LL1 yielded the co-production of 7.14 mg/L of carotenoids with 9.52 g/L of polyhydroxyalkanoates [31]. Similarly, the co-production of 2.3 mg/L of carotenoids with 4.98 g/L of polyhydroxyalkanoates was achieved by cultivating *Paracoccus* sp. LL1 in the hydrolysate of brown algae biomass [32]. The astaxanthin production of 301.14 ± 17.43 µg/g was reported in LB medium for *Paracoccus haeundaensis* [33]. In another study, Lee and Kim [34] examined the gene cluster responsible for pigment (mainly astaxanthin) production isolated from *P. haeundaensis*. They obtained about 400 µg/g of astaxanthin per dry mass from *Escherichia coli* cells transformed with a plasmid containing the studied gene cluster. Overall, the yields obtained with real waste substrates in the present study implied the viability of the scale-up process for pigment production from lignocellulosic feedstocks.

## 4. Conclusions

The present study is among the scanty reports demonstrating the possibility of utilizing lignocellulosic wastes as low-cost substrates for pigment production. The work showed that mild alkali pretreatment could be an effective approach for lignocellulosic wastes such as sawdust and wheat straw, followed by an economic saccharification operation in low energy conditions. The generated hydrolysate from both feedstocks, constituted by reducing

sugars, was found to be an attractive replacement for commercial media ingredients required for fermentation by *Paracoccus* spp. for subsequent pigment production. The comparable carotenoid yield obtained in the lignocellulosic hydrolysates, as well as in the commercial LB medium by both *P. aminophilus* CRT1 and *P. kondratievae* CRT2, was promising for probing further scale-up studies. The results of this study suggest that microbial pigment production using lignocellulosic wastes has the potential to transform the coloring and pigment industry into a healthier, more ecological, and balanced business.

**Supplementary Materials:** The following supporting information can be downloaded at: https://www.mdpi.com/article/10.3390/fermentation8090440/s1, Figure S1: *P. aminophilus* CRT1 in different liquid growth media, i.e., A (based on NaOH-pretreated wheat straw hydrolysate), B (based on NaOH-pretreated sawdust hydrolysate), C (based on microwave-assisted Ca(OH)$_2$-pretreated sawdust hydrolysate), and LB (Luria Bertani broth medium). Figure S2: *P. kondratievae* CRT2 in different liquid growth media: A (based on NaOH-pretreated wheat straw hydrolysate), B (based on NaOH-pretreated sawdust hydrolysate), C (based on microwave-assisted Ca(OH)$_2$-pretreated sawdust hydrolysate), and LB (Luria Bertani broth medium).

**Author Contributions:** Conceptualization, K.P.; investigation, W.P. and J.G.; formal analysis, W.P. and J.G.; writing—original draft preparation, K.P., W.P. and J.G.; writing—review and editing, K.P., L.D. and D.B.; supervision, K.P. All authors have read and agreed to the published version of the manuscript.

**Funding:** This research was funded by a grant from "The Fly ash as the precursors of functionalized materials for applications in environmental engineering, civil engineering and agriculture" project (no. POIR.04.04.00-00-14E6/18-00), carried out within the TEAM-NET program of the Foundation for Polish Science, co-financed by the European Union under the European Regional Development Fund and by the National Centre for Research and Development, Poland (Grant number TANGO2/339976/NCBR/2017).

**Institutional Review Board Statement:** Not applicable.

**Informed Consent Statement:** Not applicable.

**Data Availability Statement:** Not applicable.

**Conflicts of Interest:** The authors declare no conflict of interest.

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
