# Peer review of "Pigment Production by Paracoccus spp. Strains through Submerged Fermentation of Valorized Lignocellulosic Wastes"

_fermentation, doi:10.3390/fermentation8090440_

Round 1

Reviewer 1 Report

The manuscript is interesting, innovative and brings an important contribution to the field. However some important points to be considered:

-        Abstract: authors say that substrates are cost-effective for fermentation. However, no economical analysis was performed. Please change it for potential substrate.

-        Line 118: please add a space in “isolateTrichoderma sp.”

-        Item 2.3: what was biomasses’ moisture before pretreatment?

-        Line 151: why authors chose 1 g/L of initial reducing sugar? This is too low and will provide low pigment productivity. Please justify it in the text.

-        In general, 72h of fermentation is a long time to consume only 1 g/L of sugars, how authors justify the feasibility of this process?

-        Line 186 – please ensure that all microorganisms’ names in the manuscript are in italic

-        Biomass pretreatment results are not well presented. What was biomass composition before and after the pretreatments? What was the severity factor? What was the pretreatment yield? This topic needs substantial improvement.

-        Lines 251-260 – What were the inhibitors titers in hydrolysates?

-        It is necessary to perform statistical analysis in the results to ensure the superiority of certain conditions above others.

Author Response

Response to Reviewer’s Comments:

We would like to thank the Editor and the Reviewers for their constructive suggestions on our submitted manuscript. We have now incorporated the corrections and modifications suggested by the Reviewers in the revised manuscript. MS ID: 1855354; Title: Pigment production by Paracoccus spp. strains through submerged fermentation of valorized lignocellulosic wastes

Reviewer 1

Query 1:

Abstract: authors say that substrates are cost-effective for fermentation. However, no economical analysis was performed. Please change it for potential substrate.

Response:

We acknowledge the learned reviewers concern regarding potential and real time economic suitability of lignocellulosic substrates for pigment production by fermentation. In compliance, the suggested phrases have been appropriately modified in the abstract and now can be read as:

The present study aims to evaluate two such abundant lignocellulosic wastes i.e., wheat straw and pinewood saw-dust as low-cost feedstocks for carotenoid production by Paracoccus strains. An alkali pretreatment approach followed by enzymatic saccharification using an indigenous lab isolated fungal hydrolase was found to be effective for the release of fermentable sugars from both substrates. The fermentation of the pretreated sawdust hydrolysate by Paracoccus aminophilus CRT1 and Paracoccus kondratievae CRT2 resulted in highest carotenoid production of 631.33 and 758.82 μg/g dry mass respectively. Thus, the preliminary but informative research findings of the present work exhibit potency for sustainable and economically feasible pigment production from lignocellulosic feedstocks after optimal process development at the pilot scale.

Query 2:

Line 118: please add a space in “isolateTrichoderma sp.”

Response:

We appreciate the reviewer’s careful observation. The above has been rectified in the revised MS.

Query 3:

Item 2.3: what was biomasses’ moisture before pretreatment?

Response:

Both the biomass was air-dried and stored in sealed bags before being subjected to pretreatment.

Query 4:

Line 151: why authors chose 1 g/L of initial reducing sugar? This is too low and will provide low pigment productivity. Please justify it in the text.

Response:

We acknowledge the learned reviewers insightful observation. However, as justified in the text, the present study is a preliminary investigation, performed at flask level, on use of two lignocellulosic waste feedstocks without any optimizations. Further, the use of crude indigenous fungal enzyme without any concentration/purification or dosage changes and the highly recalcitrant nature of sawdust yielded ~100 mg/g reducing sugars. So, for fermentations performed in 100 ml volume, from 1g feedstock, 0.1g reducing sugar was used which translates to 1g/l (as justified in text) and for uniformity same was maintained for wheat straw hydrolysate.

Nevertheless, the authors completely acknowledge the possibility of higher pigment productivity with increased reducing sugars and further ongoing studies of research group have used solvent precipitation for concentration of hydrolases as well as other optimizations for improvement in yield of reducing sugars which might as well as lead to higher pigment productivity. However, carotenoid production is a multi-step complex biosynthetic pathway wherein assimilation of sugars as nutrient source might not completely direct the flux towards carotenoid biosynthesis and thus, inhibition of carotenoid production by Paracoocus strains in presence of high concentration of reducing sugars has been reported by many studies [Ram et al. 2020; Kumar et al. 2018].

Query 5:

In general, 72h of fermentation is a long time to consume only 1 g/L of sugars, how authors justify the feasibility of this process?

Response:

As discussed above, the authors completely acknowledge the possibility of better feasibility or pigment productivity with increased reducing sugars and accept that present study is a preliminary investigation, performed at flask level, on use of two lignocellulosic waste feedstocks without any optimizations. As stated in the text, the preliminary research findings of present study exhibit potency for sustainable and eco-nomically feasible pigment production from lignocellulosic feedstocks after optimal process development at the pilot scale.

Further, the use of 72 h fermentation time is justified as the pigments are recognized as secondary metabolites which are produced in stationary phase, wherein assimilation of most of the nutrients have already taken place [Chougle & Singhal, 2012; Ram et al. 2020; Muhammad et al. 2020].

Reference:

Chougle, J. A., & Singhal, R. S. (2012). Metabolic precursors and cofactors stimulate astaxanthin production in Paracoccus MBIC 01143. Food Science and Biotechnology21(6), 1695-1700.

Ram, S., Mitra, M., Shah, F., Tirkey, S. R., & Mishra, S. (2020). Bacteria as an alternate biofactory for carotenoid production: A review of its applications, opportunities and challenges. Journal of Functional Foods, 67, 103867.

Muhammad, M.; Aloui, H.; Khomlaem, C.; Hou, C.T.; Kim, B.S. Production of Polyhydroxyalkanoates and Carotenoids through Cultivation of Different Bacterial Strains Using Brown Algae Hydrolysate as a Carbon Source. Biocatal. Agric. Biotechnol. 2020, 30, 101852, doi:https://doi.org/10.1016/j.bcab.2020.101852.

Query 6:

Line 186 – please ensure that all microorganisms’ names in the manuscript are in italic

Response:

We apologize for the error and all microorganisms’ names are in italics in the revised MS.

Query 7:

Biomass pretreatment results are not well presented. What was biomass composition before and after the pretreatments? What was the severity factor? What was the pretreatment yield? This topic needs substantial improvement.

Response:

We highly appreciate such important comment from the reviewer and understand the critical role of pretreatment in improving yield of reducing sugars. However, withstanding the limitations of present preliminary investigative study, it was not possible to evaluate yield or optimize pretreatment conditions. Nevertheless, the biomass composition of dried samples before pretreatment was determined by standard NREL procedures and have been incorporated in the revised MS.

In context of severity factor, for alkali pretreatment performed with NaOH (pH 12) at room temperature (22 °C) for 60 minutes comes out at -12.52 calculated by standard formula:

Combined Severity Factor = log t*exp((T(t)- 100)/14.74)-pH = -0.52-12= -12.52

wherein where t is the holding time of treatment in min, T(t) is the treatment temperature, 100 is the reference temperature.

This is in agreement with severity factor reported in previous studies [Pedersen & Meyer 2010;   Fang et al. 2022] for similar temperature and pH range. However, this has not been incorporated in revised MS to prevent chaos and maintain uniformity with Ca(OH)2 pretreatment which was assisted by microwave treatment at 1000W for 6 min.

Reference:

Pedersen, M., & Meyer, A. S. (2010). Lignocellulose pretreatment severity–relating pH to biomatrix opening. New biotechnology, 27(6), 739-750.

Fang, L., Su, Y., Wang, P., Lai, C., Huang, C., Ling, Z., & Yong, Q. (2022). Co-production of xylooligosaccharides and glucose from birch sawdust by hot water pretreatment and enzymatic hydrolysis. Bioresource Technology348, 126795.

Query 8:

Lines 251-260 – What were the inhibitors titers in hydrolysates?

Response:

Inhibitory compounds in lignocellulosic hydrolysates might comprise of aliphatic acids (i.e. acetic, formic and levulinic acid), furaldehydes (i.e. furfural and 5-hydroxymethylfurfural (HMF)), aromatic compounds (i.e. phenolics), whose titre vary with the nature of biomass and

the pretreatment conditions used. The literature studies report generation of Furfural  ~0.15±0.02; acetic acid ~2.70±0.33 g/l, and HMF ~0.1 g/l from wheat straw [Chandel et al. 2013] while Furfural <0.5g/l and HMF <1g/l [Rusanen et al. 2019] are reported as main inhibitory compounds from pinewood sawdust.

However, it was not feasible to measure the exact titers of these inhibitors in present study hydrolysates and same has been incorporated in revised MS.

Reference:

Rusanen, Annu, et al. "Selective hemicellulose hydrolysis of Scots pine sawdust." Biomass Conversion and Biorefinery 9.2 (2019): 283-291.

Chandel, Anuj K., Silvio Silvério Da Silva, and Om V. Singh. "Detoxification of lignocellulose hydrolysates: biochemical and metabolic engineering toward white biotechnology." BioEnergy Research 6.1 (2013): 388-401.

Query 9:

It is necessary to perform statistical analysis in the results to ensure the superiority of certain conditions above others.

Response:

The authors acknowledge the reviewer’s suggestion, but since no optimization studies were performed, usage of statistical tools becomes limited. Nevertheless, all the experiments were performed in triplicates and the data presented are mean ± standard deviation of replicates.

Reviewer 2 Report

This paper aimed to study the carotenoids production through submerged fermentation of lignocellulosic wastes. The content mainly includes lignocellulosic wastes pretreatment and pigment fermentation. I have found some merits. However, this paper contained some information, which should be presented as preliminary experiment and not shown as main results. It still needs further investigation and sufficiently novel. Therefore, this manuscript is inappropriate for publishing in Fermentation. 

1. The source of the strains was not clearly explained. It has been mentioned that the strain used in this paper was carrying plasmid pCRT01Line 181),and Ref.19 was citedMiller, G.L. Use of dinitrosalicylic acid reagent for determination of reducing sugar. Anal. Chem. 1959 31(3), 426–428. I am not sure Ref 19 is related to this strain. Furthermoreis the engineered strain constitutive or inducible, and is it necessary to add an inducer in the fermentation process? None of them were explained in the text.

2.  Line 180Strain name needs italics. “Paracoccus spp. P. aminophilus CRT1” should be Paracoccus spp. P. aminophilus CRT1.

3. Line 284, ”As far as synthetic medium (Glucose & LB medium) was concerned” . It is not rigorous that LB medium is called synthetic medium.

4. In Fig 6 and Fig 7why total carotenoids were detected at two different wavelengths488 nm and 453 nm

5. As mentioned in 2.5,“The reducing sugar content of all the hydrolysates was set to 1g/L and supplemented with 0.5% (w/v) yeast extract.The composition of the culture medium is unreasonable. From the low OD value, the growth of the bacteria is not good. On the one hand, hydrolysis inhibitors are considered; on the other hand, the lower carbon and nitrogen source concentration is not enough to maintain the nutritional needs of 72 h fermentation.

6. It is strongly recommended to systematically optimize the fermentation medium and culture conditions by response surface methodology or statistical analysis, so that significantly improve the yield of pigment.

7. The clarity of figures used in this research needs to be improved.

Author Response

Response to Reviewer’s Comments:

We would like to thank the Editor and the Reviewers for their constructive suggestions on our submitted manuscript. We have now incorporated the corrections and modifications suggested by the Reviewers in the revised manuscript.

MS ID : 1855354

Title: Pigment production by Paracoccus spp. strains through submerged fermentation of valorized lignocellulosic wastes

This paper aimed to study the carotenoids production through submerged fermentation of lignocellulosic wastes. The content mainly includes lignocellulosic wastes pretreatment and pigment fermentation. I have found some merits. However, this paper contained some information, which should be presented as preliminary experiment and not shown as main results. It still needs further investigation and sufficiently novel. Therefore, this manuscript is inappropriate for publishing in Fermentation.

Response:

We thank the reviewer for the constructive criticism and completely agree that it is a preliminary investigative study, which needs to be supplemented with more optimizations before real time application. This has been highlighted in the revised MS at all required instances for transparency. The abstract itself states: Thus, the preliminary but informative research findings of the present work exhibit potency for sustainable and economically feasible pigment production from lignocellulosic feedstocks after optimal process development at the pilot scale. Further, both wheat straw and pinewood sawdust have not been previously investigated as  feedstocks for pigment production by Paracoccus or other related bacterial genus, which strengthens the originality of the present study in advocating lignocellulosic biomass as drivers of natural and sustainable pigments as an alternate to chemical or synthetic cascade.

Query 1:

The source of the strains was not clearly explained. It has been mentioned that the strain used in this paper was carrying plasmid pCRT01Line 181),and Ref.19 was citedMiller, G.L. Use of dinitrosalicylic acid reagent for determination of reducing sugar. Anal. Chem. 1959 31(3), 426–428. I am not sure Ref 19 is related to this strain. Furthermore is the engineered strain constitutive or inducible, and is it necessary to add an inducer in the fermentation process? None of them were explained in the text.

Response:

The authors apologize for the Ref mismatch and it stands corrected in Revised MS. The engineered strain expression is constitutive (Maj et al. 2020) as implied in the Methods section and also exhibited in Colony morphology depicted in Fig. 1.

Reference:

  1. Maj, A.; Dziewit, L.; Drewniak, L.; Garstka, M.; Krucon, T.; Piatkowska, K.; Gieczewska, K.; Czarnecki, J.; Furmanczyk, E.; 386 Lasek, R. In Vivo Creation of Plasmid PCRT01 and Its Use for the Construction of Carotenoid-Producing Paracoccus 387 Strains That Grow Efficiently on Industrial Wastes. Microb. Cell Fact. 2020, 19, 1–14

Query 2:

Line 180Strain name needs italics. “Paracoccus spp. P. aminophilus CRT1” should be Paracoccus spp. P. aminophilus CRT1.

Response:

We apologize for this typographic mistake and all microorganisms’ names are in italics in the revised MS.

Query 3:

Line 284, ”As far as synthetic medium (Glucose & LB medium) was concerned” . It is not rigorous that LB medium is called synthetic medium.

Response:

We thank the reviewer for this valuable insight. The Line 284 (As far as synthetic medium (Glucose & LB medium) was concerned) as well synthetic medium ambiguity has been removed from entire revised MS for better clarity.

Query 4:

In Fig 6 and Fig 7why total carotenoids were detected at two different wavelengths488 nm and 453 nm

Response:

Carotenoids, are one of the most diverse class of pigments and usually the microbes (including Paracoccus sp.) produce a mixture of carotenoids viz. xanthophylls (astaxanthin, adonixanthin, adonirubin and canthaxanthin) and carotenes (echinenone, hydroxyechinenone and β-carotene). It is a well-established protocol in both spectrophotometric and HPLC procedures to observe absorption peaks at this range for total carotenoids [Maj et al. 2020; Strati et al. 2012].

Table 1

Tentative identification, chromatographic data and content (μg/100g dry basis) for all-trans and cis forms of carotenoids in tomato waste.

Peak No.

Compound

RT(min)

λ (nm)Found

λ (nm) Reported

Q-Ratio Found

Q-Ratio Reported

k

α

Content (μg/100 g Dry Basis )

1

All- trans-lutein

3.37 ± 0.04

423, 447, 477

422, 446, 476

0.04

0.06

0.53

2.06

39.14 ± 0.21

2

9- cis-lutein

4.94 ± 0.06

350, 420, 442, 474

356, 428, 446, 476

0.1

0.12

1.24

1.32

17.59 ± 0.05

3

13 -cis-lutein

5.91 ±0.09

376, 437, 458, 485

374, 434, 458, 488

0.31

0.33

1.69

1.51

42.69 ± 0.03

4

All- trans-β-carotene

7.97 ± 0.03

428, 454, 482

458, 482

-

0.12

2.62

1.15

48.48 ± 0.92

5

9 -cis-β-carotene

8.90 ± 0.08

340, 449, 480

344, 452, 476

0.11

0.12

3.04

1.88

4.24 ± 0.01

6

13- cis-β-carotene

15.09 ± 0.05

345, 451, 479

344, 422, 458, 476

0.34

0.35

5.86

2.15

4.42 ± 0.01

7

All- trans-lycopene

30.23 ± 0.20

450, 476, 507

452, 476, 506

-

0.06

12.74

2.15

64.84 ± 0.87

RT: retention time; k: capacity factor; α: separation (selectivity) factor; three independent samples were analyzed; data are expressed as mean ± standard deviation (n = 3).

Table from Reference:

  • Strati, I. F., Sinanoglou, V. J., Kora, L., Miniadis-Meimaroglou, S., & Oreopoulou, V. (2012). Carotenoids from foods of plant, animal and marine origin: An efficient HPLC-DAD separation method. Foods1(1), 52-65.

However, as suggested for clarity, the total carotenoid production is now modified and presented in Figures.

Query 5:

As mentioned in 2.5The reducing sugar content of all the hydrolysates was set to 1g/L and supplemented with 0.5% (w/v) yeast extract.” The composition of the culture medium is unreasonable. From the low OD value, the growth of the bacteria is not good. On the one hand, hydrolysis inhibitors are considered; on the other hand, the lower carbon and nitrogen source concentration is not enough to maintain the nutritional needs of 72 h fermentation.

Response:

We acknowledge the learned reviewers insightful observation. However, as justified in the text, the present study is a preliminary investigation, performed at flask level, on use of two lignocellulosic waste feedstocks without any optimizations. Further, the use of crude indigenous fungal enzyme without any concentration/purification or dosage changes and the highly recalcitrant nature of sawdust yielded ~100 mg/g reducing sugars. So, for fermentations performed in 100 ml volume, from 1g feedstock, 0.1g reducing sugar was used which translates to 1g/l (as justified in text) and for uniformity same was maintained for wheat straw hydrolysate.

The OD value >1.4 (Figure 5) supports luxuriant growth of bacteria in all the media.

Further, the use of 72 h fermentation time is justified as the pigments are recognized as secondary metabolites which are produced in stationary phase, wherein assimilation of most of the nutrients have already taken place [Chougle & Singhal, 2012; Ram et al. 2020; Muhammad et al. 2020].

Reference:

  • Chougle, J. A., & Singhal, R. S. (2012). Metabolic precursors and cofactors stimulate astaxanthin production in Paracoccus MBIC 01143. Food Science and Biotechnology, 21(6), 1695-1700.
  • Ram, S., Mitra, M., Shah, F., Tirkey, S. R., & Mishra, S. (2020). Bacteria as an alternate biofactory for carotenoid production: A review of its applications, opportunities and challenges. Journal of Functional Foods, 67, 103867.
  • Muhammad, M.; Aloui, H.; Khomlaem, C.; Hou, C.T.; Kim, B.S. Production of Polyhydroxyalkanoates and Carotenoids through Cultivation of Different Bacterial Strains Using Brown Algae Hydrolysate as a Carbon Source. Biocatal. Agric. Biotechnol. 2020, 30, 101852, doi:https://doi.org/10.1016/j.bcab.2020.101852.

Query 6:

It is strongly recommended to systematically optimize the fermentation medium and culture conditions by response surface methodology or statistical analysis, so that significantly improve the yield of pigment.

Response:

The authors completely acknowledge the possibility of higher pigment productivity with statistical optimization and further ongoing studies of research group have used solvent precipitation for concentration of hydrolases as well as other optimizations for improvement in yield of reducing sugars, which might as well as lead to higher pigment productivity in future. However, in current scenario, the authors request to understand their limitation in providing detailed optimizations to the presented investigative preliminary study.

Query 7:

The clarity of figures used in this research needs to be improved.

Response:

The clarity of figures has been improved (600dpi) for succinct representation of results with easy comprehension.

Reviewer 3 Report

The authors are trying to produce pigment from lignocellulose using Paracoccus spp. In particular, the authors are trying to produce carotenoid pigments from Paracoccus spp.

The authors' paper does not clearly state the reason for producing pigment from lignocellulose using microorganisms. Also, the authors do not provide any references on microbial pigment production. In particular, the authors do not explain the background of the research on carotenoid pigment production by Paracoccus spp. These are very important contents in showing the originality of the authors' research.

Authors need to explain these things. Also, the authors specifically use two strains.

The authors should also state the reason for this.

The authors show that Paracoccus aminophilus grows better than Paracoccus kondratievae. On the other hand, carotenoid production from lignocellulose hydrolyzate indicates that P. kondratievae is highly productive. Carotenoid production from lignocellulose hydrolysates appears to be more practical in P. kondratievae than in P. aminophilus. The authors should state the reason for this.

In addition, the authors need to show the results of comparative analysis of the results of fermentative production from lignocellulose in this paper and the fermentative production of carotenoid pigments by common Paracoccus spp. This can be very important to the originality of this paper.

Data for Supplementary Figure S1 and Figure S2 are not available.

Author Response

Response to Reviewer’s Comments:

We would like to thank the Editor and the Reviewers for their constructive suggestions on our submitted manuscript. We have now incorporated the corrections and modifications suggested by the Reviewers in the revised manuscript.

MS ID : 1855354

Title: Pigment production by Paracoccus spp. strains through submerged fermentation of valorized lignocellulosic wastes

 Reviewer 2

Query 1:

The authors' paper does not clearly state the reason for producing pigment from lignocellulose using microorganisms. Also, the authors do not provide any references on microbial pigment production. In particular, the authors do not explain the background of the research on carotenoid pigment production by Paracoccus spp. These are very important contents in showing the originality of the authors' research. Authors need to explain these things.

Response:

We are thankful for the reviewer’s observation. In context to the utilization of lignocellulosic biomass for microbial pigment products, the authors group review [Grewal et al. 2022] and several other recent studies [de Medeiros 2022, Lopes 2021] have emphasized the utilization of ~100 X 108 metric tons of lignocellulosic biomass produced annually as a sustainable alternate strategy to chemically synthesized pigments with multiple benefits of waste valorisation, environmental protection and generation of high value product. Majority of the agro-industrial wastes are lignocellulosic in nature and their valorisation as feedstocks for pigment production will result in generation of economically viable and safe pigments, after overcoming the bottlenecks in their real time industrial production. This has been duly emphasized in the revised MS and the recent references on microbial pigment production from lignocellulosic wastes have also been incorporated.

References:

de Medeiros, T.D.M.; Dufossé, L.; Bicas, J.L. Lignocellulosic Substrates as Starting Materials for the Production of Bioactive Biopigments. Food Chem. X 2022, 13, 100223.

Lopes, F.C.; Ligabue-Braun, R. Agro-Industrial Residues: Eco-Friendly and Inexpensive Substrates for Microbial Pigments Production. Front. Sustain. Food Syst. 2021, 5, 589414.

Grewal, J.; Woła̧cewicz, M.; Pyter, W.; Joshi, N.; Drewniak, L.; Pranaw, K. Colorful Treasure From Agro-Industrial Wastes: A Sustainable Chassis for Microbial Pigment Production. Front. Microbiol. 2022, 13, 832918.

Muhammad, M.; Aloui, H.; Khomlaem, C.; Hou, C.T.; Kim, B.S. Production of Polyhydroxyalkanoates and Carotenoids through Cultivation of Different Bacterial Strains Using Brown Algae Hydrolysate as a Carbon Source. Biocatal. Agric. Biotechnol. 2020, 30, 101852, doi:https://doi.org/10.1016/j.bcab.2020.101852.

Maj, A.; Dziewit, L.; Drewniak, L.; Garstka, M.; Krucon, T.; Piatkowska, K.; Gieczewska, K.; Czarnecki, J.; Furmanczyk, E.; Lasek, R. In Vivo Creation of Plasmid PCRT01 and Its Use for the Construction of Carotenoid-Producing Paracoccus Spp. Strains That Grow Efficiently on Industrial Wastes. Microb. Cell Fact. 2020, 19, 1–14.

For better comprehension of the background on the carotenoid pigment production by Paracoccus spp., the details below are provided and also appropriately added in the revised MS.

Carotenoid pigments due to their bright coloration, structural diversity and bioactivity have garnered high nutritional and pharmocological interest with estimated market value of 2 billion dollars by 2027 [5]. However, chemical routes drives the synthesis of 80-90% of commercially available carotenoids. Hence, in sync with consumer demand and better acceptance for natural carotenoids, the microbial sources are gaining attention and genus Dietzia and Paracoccus are reported as attractive candidates [10,11].

Further, the previous work of authors [Maj et al. 2020] also proved that used Paracoccus spp. due to their genetic background were highly robust and exhibited growth in many complex media as well as raw industrial effluent (coal-fired power plant fuel gas desulfurization wastewater), which strengthened their potency for industrial or biotechnological applications under harsh conditions. Thus, validating the above observations about the two Paracoccus spp., the present study leveraged the metabolic machinery of these two strains for pigment production via utilization of waste lignocellulosic substrates which due to their structural recalcitrance aren’t easily amenable for fermentation technology. Both wheat straw and pinewood sawdust have not been previously investigated as  feedstocks for pigment production by Paracoccus or other related bacterial genus, which justifies the originality of the present research.

Query 2:

Also, the authors specifically use two strains. The authors should also state the reason for this.

Response:

We are thankful for the reviewer’s observation. As discussed above, various Paracoccus species viz. Paracoccus haeundaensis, Paracoccus carotinifaciens, Paracoccus marcusii have been reported as carotenoid producers. The previous work of the authors [Maj et al. 2020] used in vivo creation of plasmids to transform colorless strains of Paracoccus sp. into efficient pigment producers. The transformed strains, i.e., Paracoccus aminophilus CRT1 and Paracoccus kondratievae CRT2 could effectively grow on industrial effluents, i.e., flue gas desulfurization (FGD) wastewater supplemented with molasses and produced carotenoids. On similar lines, the present study was based on the rationale of harnessing robustness of these strains for pigment production from unconventional but waste substrates as an alternative to pure but expensive synthetic nutrient sources for fermentation. Therefore, in chronology and for strengthened validation, the pigment production was undertaken with two strains to establish feasibility of a socially, economically and environmentally friendly bioprocess from abundant waste substrates.

Query 3:

The authors show that Paracoccus aminophilus grows better than Paracoccus kondratievae. On the other hand, carotenoid production from lignocellulose hydrolyzate indicates that P. kondratievae is highly productive. Carotenoid production from lignocellulose hydrolysates appears to be more practical in P. kondratievae than in P. aminophilus. The authors should state the reason for this.

Response:

We agree with learned reviewer observation that P. kondratievae might be more conducive for pigment production from lignocellulosic hydrolysates. The same has been incorporated in revised MS.

In comparison, P. kondratievae CRT2 proved to be a more promising strain for utilization of lignocellulosic feedstocks for pigment production (Figure 7). The highest carotenoid production (758.82 µg/g of dry mass) was observed when grown in the medium containing NaOH pretreated sawdust hydrolysate, which was even higher than the titre obtained in synthetic medium LB (748.66 µg/g).

However, in context to the point regarding comparative growth of these two strains, we would like to state that the two strains did not exhibit a very stark difference in OD600nm while growing in lignocellulosic hydrolysates i.e. The growth of P. aminophilus CRT1 in the NaOH and microwave-assisted Ca(OH)2 pretreated SD hydrolysate medium was observed as OD600 > 1.81 and OD600>1.78, respectively, whereas the growth of P. kondratievae CRT2 was OD600 > 1.65 and OD600 > 1.60 respectively.

Further, carotenoid production  is a multi-step complex biosynthetic pathway wherein assimilation of sugars as nutrient source might not completely direct the flux towards carotenoid biosynthesis. This variation in amount of carotenoid produced with diverse Paracoocus strains and differently assimilated carbon sources has been reported by many studies [Kumar et al. 2018; Chougle & Singhal, 2012; Ram et al. 2020]. For instance for Paracoccus MBIC 01143, Chougle & Singhal, (2012) reported that Sugars such as sucrose, maltose, lactose, fructose, and maltodextrin supported growth but not astaxanthin formation, whereas starch and galactose yielded poor growth as well as astaxanthin.

Reference:

Chougle, J. A., & Singhal, R. S. (2012). Metabolic precursors and cofactors stimulate astaxanthin production in Paracoccus MBIC 01143. Food Science and Biotechnology21(6), 1695-1700.

Kumar, P.; Jun, H.-B.; Kim, B.S. Co-Production of Polyhydroxyalkanoates and Carotenoids through Bioconversion of Glycerol by Paracoccus Sp. Strain LL1. Int. J. Biol. Macromol. 2018, 107, 2552–2558

Ram, S., Mitra, M., Shah, F., Tirkey, S. R., & Mishra, S. (2020). Bacteria as an alternate biofactory for carotenoid production: A review of its applications, opportunities and challenges. Journal of Functional Foods, 67, 103867.

Query 4:

In addition, the authors need to show the results of comparative analysis of the results of fermentative production from lignocellulose in this paper and the fermentative production of carotenoid pigments by common Paracoccus spp. This can be very important to the originality of this paper.

Response:

The authors are highly grateful for the reviewer’s suggestion. In compliance with reviewers’ suggestion, the following details have been provided in the revised MS.

The studies evaluating waste utilization for pigment production using the Paracoccus strains or its gene cluster are scanty. Nevertheless, the yields obtained in the present study were comparable or higher than commonly reported titers for Paracoccus strains in previous studies. The carotenoid production using flue gas desulfurization wastewater (FGD) supplemented with molasses as feedstock by P. aminophilus and P. kondratievae was reported as 117.68 and 60 µg/g of dry mass respectively [11]. The utilization of glycerol, an industrial effluent as cost-effective substrate by Paracoccus sp. LL1 yielded co-production of 7.14 mg/L carotenoids with 9.52 g/L polyhydroxyalkanoates [29]. Similarly, the co-production of 2.3 mg/ L carotenoids with 4.98 g/ L polyhydroxyalkanoates was achieved by cultivating Paracoccus sp. LL1 in hydrolysate of brown algae biomass [30].The astaxanthin production of 301.14 ± 17.43 µg/g was reported in LB medium for Paracoccus haeundaensis [31]. In another study, Lee and Kim [32] examined the gene cluster responsible for pigment (mainly astaxanthin) production isolated from P. haeundaensis. They obtained about 400 µg/g of astaxanthin per dry mass from Escherichia coli cells transformed with a plasmid containing the studied gene cluster.

Nevertheless, it is important to bring to attention that it is difficult to provide a fair comparative analysis of carotenoid pigment titers by Paracoccus spp. or other related strains from waste feedstocks due to the following variables: (i) variation in lignocellulosic composition i.e. cellulose, hemicellulose, lignin of the raw feedstock used (ii) type of pretreatment method used viz. Physical, chemical or biological along with reaction conditions (iii) source and dosage of hydolysing enzymes used for generation of fermentable sugars (iv) heterogeneity of pigment titre units (mg/g, OD units/g, mg/L, U/mL, AU/L) used in different studies with diverse feedstocks (v) variation in content of mixture of carotenoids produced with respect to xanthophylls (astaxanthin, adonixanthin, adonirubin and canthaxanthin) and carotenes (echinenone, hydroxyechinenone and β-carotene)

Query 5:

Data for Supplementary Figure S1 and Figure S2 are not available.

Response:

The authors apologize for the error and it has been rectified by attachment of Supplementary Figure S1 and Figure S2 at the end of the manuscript.

Round 2

Reviewer 1 Report

The authors have attended all the corrections accordingly

Author Response

Dear Reviewer,

On behalf of all authors, I would like to thank you for your constructive comments, which make the article more appropriate.

Thanks and regards

Kumar

Reviewer 2 Report

Although the author has made some modifications, no more convincing data and conclusions have been added. The contents presented are still preliminary experimental results, which are not innovative in terms of lignocellulosic wastes pretreatment method or pigment yield.

Author Response

Dear Reviewer,

We would like to thank you for the constructive criticism and completely agree that it is a preliminary investigative study, which needs to be supplemented with more optimizations before real-time application. Regarding the novelty of the research, as you know saccharified sawdust is having several limitations for further utilization due to the presence of phenolic compounds, and that's why not been explored much for several value-added products. In context with microbial pigments, both wheat straw and pinewood sawdust have not been previously investigated as feedstocks for pigment production by Paracoccus or another related bacterial genus, which strengthens the originality of the present study in advocating lignocellulosic biomass as drivers of natural and sustainable pigments as an alternative to chemical or synthetic cascade.

Hopefully, I have answered your query.

Thanks and Regards

Kumar

Reviewer 3 Report

The authors have made appropriate corrections.

Do Figures S1 and S2 supplement the results of Figures 5, 6, and 7?

The authors must correct the data in Figures S1 and S2.

This data has very poor resolution. For this reason, the meaning of this data cannot be understood. The authors need to recreate this data. The authors should add data for uninoculated medium only. Also, the authors need specific culture medium photo data including this data. This is important for the reliability and concreteness of the data.

Author Response

Dear Reviewer,

We would like to thank the Reviewer for the constructive suggestions on our submitted manuscript.  Please find below the response for the different queries.

Query 1: 

Do Figures S1 and S2 supplement the results of Figures 5, 6, and 7?

Response: 

Yeah, Figures S1 and S2 supplement the results of Figures 5, 6, and 7 in real conditions. 

Query 2:

The authors must correct the data in Figures S1 and S2. This data has very poor resolution. For this reason, the meaning of this data cannot be understood. The authors need to recreate this data. 

Response:  

The resolution of Fig. S1 & S2 is enhanced but removed from the manuscript word file because the JPGE file loses the resolution when pasted over the word. That's why it is now uploaded (600 dpi image) under the supplementary file option, which can be downloaded and checked separately.

Query 3:

The authors should add data for uninoculated mediums only. Also, the authors need specific culture medium photo data including this data. This is important for the reliability and concreteness of the data.

Response:

All the data were collected with different mediums (different reducing sugar in minimal medium or LB) during this study, always an uninoculated control was used. All uninoculated medium were pale yellow in color only, but we have not had any photos of that now, So incorporating a current image into the previous data is not feasible at this stage. 

Thanks and regards

Kumar